Forkhead-associated phosphopeptide binding domain 1 (FHAD1) deficiency impaired murine sperm motility

Zhang Xi 1 2
Xue Jiangyang 2 3
Jiang Shan 4
Zheng Haoyu 2 5 zhenghaoyu89@126.com
Wang Chang 2 4 wangchang@ahtcm.edu.cn
1 Department of Reproductive Health and Infertility Clinic, The Affiliated Huai’an No. 1 People’s Hospital of Nanjing Medical University , Huai’an, Jiangsu , China
2 Department of Histology and Embryology, Nanjing Medical University , Nanjing, Jiangsu , China
3 The Central Laboratory of Birth Defects Prevention and Control, Ningbo Key Laboratory for the Prevention and Treatment of Embryogenic Diseases, Women and Children’s Hospital of Ningbo University , Ningbo, Zhejiang , China
4 College of Nursing, Anhui University of Chinese Medicine , Hefei, Anhui , China
5 Department of Gynaecology, The Affiliated Huai’an No. 1 People’s Hospital of Nanjing Medical University , Huai’an, Jiangsu , China
Uversky Vladimir
Electronic publication date: 2024 Mar 29
Publication date: 2024
Volume: 12
Electronic Location ID: e17142
Received 2023 Dec 20; Accepted 2024 Feb 29
Copyright: © 2024 Zhang et al.
Copyright year: 2024
Copyright holder: Zhang et al.
License: This is an open access article distributed under the terms of the Creative Commons Attribution License, which permits unrestricted use, distribution, reproduction and adaptation in any medium and for any purpose provided that it is properly attributed. For attribution, the original author(s), title, publication source (PeerJ) and either DOI or URL of the article must be cited.
License URL: https://creativecommons.org/licenses/by/4.0/

Keywords: Testis, Spermatogenesis, Knockout, Male fertility, FHAD1

Funding: Nanjing Medical Univertisy NMUB20220214 to Xi Zhang Universities of Anhui Province Education Department 2023AH050843 Natural Science Foundation of Huai’an HAB202305 Anhui University of Chinese Medicine 2023rcyb022 The work was supported by the Science and Technology Development Foundation of Nanjing Medical Univertisy (NMUB20220214 to Xi Zhang), the Key Project of Natural Science Foundation for Universities of Anhui Province Education Department (2023AH050843 to Chang Wang), the Natural Science Foundation of Huai’an (HAB202305 to Haoyu Zheng), and the exceptional support plan of talent introduction of Anhui University of Chinese Medicine (2023rcyb022 to Chang Wang). The funders had no role in study design, data collection and analysis, decision to publish, or preparation of the manuscript.

==============================
Background

Genetic knockout-based studies conducted in mice provide a powerful means of assessing the significance of a gene for fertility. Forkhead-associated phosphopeptide binding domain 1 (FHAD1) contains a conserved FHA domain, that is present in many proteins with phospho-threonine reader activity. How FHAD1 functions in male fertility, however, remains uncertain.

Methods

Fhad1−/− mice were generated by CRISPR/Cas9-mediated knockout, after which qPCR was used to evaluate changes in gene expression, with subsequent analyses of spermatogenesis and fertility. The testis phenotypes were also examined using immunofluorescence and histological staining, while sperm concentrations and motility were quantified via computer-aided sperm analysis. Cellular apoptosis was assessed using a TUNEL staining assay.

Results

The Fhad1−/−mice did not exhibit any abnormal changes in fertility or testicular morphology compared to wild-type littermates. Histological analyses confirmed that the testicular morphology of both Fhad1−/−and Fhad1+/+ mice was normal, with both exhibiting intact seminiferous tubules. Relative to Fhad1+/+ mice, however, Fhad1−/−did exhibit reductions in the total and progressive motility of epididymal sperm. Analyses of meiotic division in Fhad1−/−mice also revealed higher levels of apoptotic death during the first wave of spermatogenesis.

Discussion

The findings suggest that FHAD1 is involved in both meiosis and the modulation of sperm motility.

Introduction

Infertility affects approximately 10‒15% of couples throughout the world, with males and females being impacted at roughly equal rates (Cooke & Saunders, 2002). Spermatogenesis is the process that gives rise to mammalian male germ cells, with spermatogonia initially undergoing meiotic division and proliferation, followed by homologous chromosome separation. During spermiogenesis, round spermatids transform into long spermatids, followed by detachment from the seminiferous epithelium (Hess & Renato de Franca, 2008). The spermatogenic process is complex, relying on control mediated by numerous genes (Oyama et al., 2022). While the expression of approximately 2,000 genes to date is known to be elevated in the testis (Schultz, Hamra & Garbers, 2003), most have yet to undergo functional verification and characterization.

Gene-editing efforts, particularly those facilitated by CRISPR/Cas9 systems, have emerged as an effective means of knocking out particular genes of interest, including those showing enrichment in testicular tissue, thereby enabling detailed analyses of their functional roles (Abbasi, Miyata & Ikawa, 2018; Jiang et al., 2014). These and related approaches have identified many key genes associated with male fertility, including Fbxo47, Tcte1, Majin, Dnali1, and Meiob (Castaneda et al., 2017; Hua et al., 2019; Rashid et al., 2006; Shibuya et al., 2015; Souquet et al., 2013). In contrast, other genes including Ttll3, Ttll8, and Cct6b, have also been found to play important regulatory roles in spermatogenesis, even though their knockout does not result in male infertility (Gadadhar et al., 2021; Yang et al., 2021). As such, there is a clear need for further studies aimed at identifying other key proteins involved in the spermatogenic process to better clarify the mechanisms that govern male fertility.

Phosphorylation is vital for the regulation of sperm-related processes required for fertilization, including sperm motility, capacitation, and the acrosome reaction (Serrano, Garcia-Marin & Bragado, 2022). The most common phosphorylation sites in mammalian proteins are serine and threonine residues, and many protein families capable of recognizing phosphorylated versions of these residues (pSer/pThr) have been identified to date (Yaffe & Elia, 2001). These include proteins harboring forkhead-associated (FHA) domains, which specifically recognize pSer/pThr residues (Durocher et al., 2000).

Many proteins associated with the chromosomes contain FHA domains, which facilitate interactions between protein modules based on specific pSer/pThr recognition (Durocher et al., 1999; Durocher & Jackson, 2002; Mohammad & Yaffe, 2009; Reinhardt & Yaffe, 2013). FHA domains are involved in various processes, such as DNA repair, protein degradation, signal transduction, protein transport, and transcription (Durocher & Jackson, 2002). The present study focused specifically on the protein FHA phosphopeptide binding domain 1 (FHAD1), which in mice is a 1,420 amino acid protein encoded by the Fhad1 gene on chromosome 4. This gene was selected as it is enriched in the testis, as shown in the NCBI database (Yue et al., 2014).

For this study, a CRISPR/Cas9 strategy was used to introduce a 28-bp shift mutation within the exon 3 of the murine Fhad1 gene, after which the reproductive consequences of Fhad1 knockout were evaluated. The subsequent analyses established FHAD1 as functioning as a sperm motility regulator in mice.

Materials and Methods

Animal studies

All mice were obtained from and maintained under specific pathogen-free (SPF) conditions in the Laboratory Animal Center of Nanjing Medical University. The mice were maintained in a pathogen-free animal housing (20–22 °C, 50–70% humidity, 12 h light/dark cycle) with free food and water access. All mice were treated humanely and all efforts were made to minimize suffering. When appropriate, mice were euthanized with CO2 and cervical dislocation prior to tissue sample collection. Each ventilated cage housed 4–5 mice, and cages were routinely cleaned, with bedding being added at a consistent density. On study completion, all mice were euthanized with CO2. There were no surviving animals at the end of the study.

Animal ethics

The following information was supplied relating to ethical approvals (i.e., approving body and any reference numbers):

The use of the mice and their treatment were checked and authorized under the Animal Ethical and Welfare Committee (Approval No. IACUC-1601117). Protocols for mouse feeding and use are under the guiding principles of the Institutional Animal Care and Use Committee (IACUC) of Nanjing Medical University.

Fhad1 −/− mouse production

Fhad1−/− mice were generated using CRISPR/Cas9 as in a prior report (Shen et al., 2013). To knock out the Fhad1 gene, a pair of guide RNAs (gRNA 1: 5′-gtgcagagaagttggtcacaggg-3′ and gRNA 2: 5′- gtgaggggcaggtgggaccatgg-3′) were designed onto the exon3 of the Fhad1 gene (NC_000070). Cas9 and sgRNA plasmids were linearized with AgeI and DraI, followed by purification with the MinElute PCR Purification Kit (Qiagen, Hilden, Germany). These sequences were then respectively prepared using the MESSAGE mMACHINE T7 Ultra Kit (Ambion, Austin, TX, USA) and the MEGA Shortscript and Clear Kit (Ambion, Austin, TX, USA). Male C57BL/6N mice and superovulated wildtype females were mated to generate zygotes into which the Cas9 mRNA and sgRNA constructs were injected. The injected embryos were then transferred into the uteri of recipient pseudo-pregnant females. Sanger sequencing was used to identify the genotype of the founder mouse and its offspring.

Genotyping

To mitigate the potential for off-target phenotypes, Fhad1+/+ mice were mated with genome-edited founder mice lacking Fhad1 expression for a minimum of three generations. PCR amplification and Sanger sequencing was employed for the genotyping of all offspring with the primers detailed in Table S1.

qPCR

TRIzol (Vazyme, Xuanwu Qu, China) was used according to the provided instructions to extract tissue RNA, after which cDNA was generated with the PrimeScript Reverse Transcription Mix (Vazyme, Xuanwu Qu, China). A 7,500 real-time PCR instrument (Applied Biosystems, Waltham, MA, USA) and SYBR Green (Q131; Vazyme, Xuanwu Qu, China) were utilized for qPCR analyses, with individual 20 µL reactions containing 1 μL of cDNA, 0.8 μL of each primer, 7.4 μL of sterile water and 10 μL of SYBR Green Master Mix. The thermocycler settings were 95 °C for 5 min, followed by 40 cycles of alternating 95 °C and 60 °C steps. Melt curves were additionally generated to confirm the presence of only a single peak for each primer pair. Relative expression was assessed with the 2−ΔΔCt method, using 18S rRNA as a comparison. Primers used for these analyses are documented in Table S1.

Cell lines and reagents

This study utilized the TM4 and 293T cell lines. Both TM4 and 293T cells were obtained from the Chinese Academy of Science Cell Bank (Shanghai, China). TM4 cells were grown in high-glucose DMEM (Invitrogen, Waltham, MA, USA) with 5% FBS (HyClone, Logan, UT, USA), while 293T cells were grown in RPMI-1640 (Invitrogen, Waltham, MA, USA) with 10% FBS (HyClone, Logan, UT, USA). Both media contained 1% antibiotic-antimycotic, and 10 mg/mL gentamicin (Life Technologies, Carlsbad, CA, USA). Cells were grown at 37 °C and 5% CO2. The spermatogonial stem cell (SSC) cDNA used for Fhad1 detect was a gift from Prof. Zheng’s lab (Shen et al., 2019).

Cell sorting

A modified version of a previously published approach was used to purify germ cells (Bastos et al., 2005). Briefly, after collecting testis tissue from 8-week-old mice, the tissue samples were digested for 15 min using 1 mg/mL type IV collagenase (17104-019; Invitrogen, Waltham, MA, USA), digested for 10 min using 0.25% trypsin/1 mM EDTA (25200-072; Gibco, Waltham, MA USA), and suspended and stained for 30 min with PBS containing 0.1% Hoechst 33342 at 37 °C to enable the identification of haploid, diploid, and tetraploid cells. After three washes with PBS, the cells were sorted using a FACSAria Fusion SORP instrument (Becton Dickinson, Franklin Lakes, NJ, USA).

Analyses of fertility

Adult male Fhad1+/+ and Fhad1−/− mice were caged separately for 16 weeks with 2 Fhad1+/+ C57BL/6N females. Data on vaginal plugs, number of pups, and birth dates of litters were logged to assess fertility-related outcomes. The knockout male mice were subjected to the same feeding conditions as controls. All mice used for the fertility test were 8–12 weeks old.

Histological staining

Epididymal and testicular tissue samples were collected from a minimum of three mice per genotype, fixed for 24 h in modified Davidson’s fluid, and stored in 70% EtOH. An EtOH gradient was used for sample dehydration, after which they were paraffin-embedded, cut to produce 5 μm sections, and mounted on glass slides. Before histological examination, samples were deparaffinized and underwent hematoxylin and eosin (H&E) staining. Sperm were spread on glass slides and air-dried, after which they were fixed in 4% paraformaldehyde (PFA) at room temperature for 30 min, followed by H&E staining and morphological observation.

Computer-assisted sperm analysis

Analyses of sperm characteristics were conducted as described (Castaneda et al., 2017). For each mouse, the distal cauda region of the right epididymis was clamped, excised, and rinsed with warmed PBS, before placing in an Eppendorf tube with 200 μl of fresh human tubal fluid (HTF) media (Millipore, Burlington, MA, USA) containing 10% fetal bovine serum (FBS) that had been pre-warmed to 37 °C. After removing the clamps, the cauda was pierced with a scalpel, and sperm were able to diffuse into the surrounding media for 5 min during which they were incubated at 37 °C, after which 10 μl of the sperm suspension was used to assess sperm motility using computer-assisted sperm analysis (CASA) (Hamilton Thorne Research Inc., Beverly, MA, USA).

Immunofluorescence

Tissue sections were deparaffinized, rehydrated, washed three times using PBS (10 min per wash), and boiled in a microwave in 10 mM citrate buffer solution (pH 6.0) for antigen retrieval. After three 10-min washes with PBS, samples were blocked with blocking buffer (1% BSA in PBS with 0.1% Triton X-100) for 2 h, followed by overnight incubation with the primary anti-γH2AX antibody (ab81299, 1:1,000; Abcam, Cambridge, UK) at 4 °C. To test sperm samples, the sperm were spread directly on slides and allowed to air-dry. The sperm samples were then fixed in 4% PFA for 30 min, washed three times with PBS, blocked with 1% BSA in PBS containing 0.1% Triton X-100, and then incubated with anti-AC-TUBULIN (FNab00082, 1:1,000; FineTest, Palm Coast, FL, USA) at 4 °C overnight. Cell lines were grown on chamber slides precoated with fibronectin (FC010; Sigma, Burlington, MA, USA). The cells were washed three times with PBS and then fixed with 4% PFA for 15 min; after three times washes with PBS, the samples were blocked with 1% BSA in PBS containing 0.1% Triton X-100 and then incubated with anti-γH2AX (ab81299, 1:1,000; Abcam, Cambridge, United Kingdom) at 4 °C overnight. Secondary antibody staining was performed for 2 h at room temperature. Nuclear counterstaining was performed for 5 min using Hoechst 33342. Sections were then washed with PBS, mounted with VectaShield or Immu-Mount, and imaged with an LSM800 confocal microscope (Carl Zeiss AG, Oberkochen, Germany).

Transmission electron microscopy

Transmission electron microscopy (TEM) analysis was performed in accordance with previous reports (Wu et al., 2023). The samples were first fixed in 1% osmium tetroxide, dehydrated in a series of ethanol solutions (50%, 70%, 90%, and 100%), followed by dehydration in 100% acetone. After infiltration with acetone and SPI-Chem resin and embedding via Epon812, the samples were sectioned using an ultra-microtome and stained with uranyl acetate and lead citrate. Visualization and image capture were conducted using a Talos L120C G2 electron microscope (Thermo Fisher, Waltham, MA, USA).

TUNEL staining

An In Situ Cell Death Detection Kit (Roche Diagnostics,11684795910, Indianapolis, IN, USA) and a TUNEL Bright-Green Apoptosis Detection Kit (A112-03; Vazyme, Jiangsu, China) were used to perform TUNEL staining, as reported previously (Hua et al., 2019).

Statistical analyses

Data were analyzed with GraphPad Prism 9.0. Results are presented as the mean ± standard deviation, and were compared with two-tailed Student’s t-tests. *P < 0.05; **P < 0.01; ***P < 0.001; ****P < 0.0001.

Results

Analyses of time- and tissue-dependent patterns of murine Fhad1 expression

To begin exploring the possible functional roles that FHAD1 plays in male fertility, Fhad1 expression patterns in murine tissues were assessed using the NCBI database (Yue et al., 2014). Although Fhad1 was also expressed in small amounts in the lungs, it was highly enriched in the testes compared to other tissues. Subsequent qPCR-based analyses of mice confirmed that Fhad1 was expressed at high levels in testis samples, whereas it was largely undetectable in other tissue types except the lungs (Fig. 1A). Longitudinal analyses of the Fhad1 mRNA levels in wild-type male mice revealed that it underwent progressive upregulation from a postnatal age of 2 weeks, when the most advanced germ cells are beyond the zygotene stage (Fig. 1B). Next, fluorescence-activated cell sorting was used to purify germ cells with different ploidy from wild-type mouse testes. In addition, the qPCR analysis showed high levels of Fhad1 mRNA in both haploid and tetraploid cells, indicating the expression of Fhad1 in primary spermatocytes and spermatids (Fig. 1C). The specific expression of Fhad1 in the testis may suggest that it plays key roles in the spermatogenic process after meiotic initiation.

Figure 1 The expression pattern of Fhad1 and knockout.

(A) The expression of Fhad1 in different murine tissues (10 weeks) was assessed via qPCR n = 8. He, heart; Li, liver; Sp, spleen; Lu, lung; Ki, kidney; Br, brain; Mu, muscle; Fa, fat; St, stomach; Gu, gut; Te, testes; Ut, uterus; Ov, ovary. (B) The expression of Fhad1 in murine testes at different developmental stages, including 0, 1, 2, 3, 4, 5, 6, and 8 weeks, was assessed via qPCR. 18s served as a normalization control for all qPCR analyses n = 3. (C) The Fhad1 levels in indicated cells n = 3. (D) Schematic overview of the utilized CRISPR/Cas9 targeting approach. (E) Sanger sequencing identification of mouse genotypes, with the primers shown in the Table S1. The location of primers is shown in the D. (F) A qPCR analysis confirmed the presence of reduced Fhad1 expression in the generated Fhad1−/−mice n = 12, ****P < 0.0001.

Fhad1 −/− mouse development

To test the functional role that FHAD1 plays in male fertility, a CRISPR/Cas9 strategy was utilized to knock out the Fhad1 gene in C57BL/6 mice, yielding a stable mutant mouse line in which 28 nucleotides were deleted (Fig. 1D). Sanger sequencing was used to confirm the deletion of this DNA fragment (Fig. 1E), and qPCR analyses revealed the reduced expression of Fhad1 in these Fhad1−/− mice (Fig. 1F). However, no changes in the viability, growth, or behavior of Fhad1−/− mice were noted, thus confirming the successful establishment of a mouse line in which this gene of interest had been deleted.

Fhad1 −/− mice exhibit normal fertility phenotypes

To assess the fertility of male Fhad1−/− mice, they were housed individually with Fhad1+/+ females for at least 16 weeks, and the average litter size and number of pups were assessed for each of the males (Fig. 2A). No differences in these fertility-related parameters were observed when comparing Fhad1+/+ and Fhad1−/− males, nor did the testis sizes differ between these groups (Figs. 2B and 2C). The loss of Fhad1 expression was thus not associated with reduced male fertility.

Figure 2 Fhad1−/−testes exhibit normal spermatogenesis.

(A) The average numbers of Fhad1+/+ and Fhad1−/− pups per litter, n = 11 vs 10. (B) Testicular tissue samples from adult Fhad1+/+ and Fhad1−/− mice. (C) Average testis weight values, with body weight being used for normalization, n = 6. (D) H&E staining was used to evaluate testes sections from Fhad1+/+ and Fhad1−/− mice. (E) Relative composition ratios for different cell types in each testis section at different spermatogenic stages: VIII, n = 6, P-L, pre-leptotene; P, pachytene; RS, round spermatids. (F) Sections from Fhad1+/+ and Fhad1−/− mice were subjected to immunofluorescent staining using anti-γH2AX. The γH2AX signal in the spermatocytes and elongating spermatids were shown. (G) The ultrastructure of testicular elongating spermatids. M, midpiece; P, principal piece; ODF, outer dense fibers; FS, fibrous sheath.

Spermatogenesis occurs in a unidirectional manner, with germ cells proceeding through developmental stages that are subject to strict temporal regulation. In the testes, the level of spermatid development can be used to classify the seminiferous epithelial cycle into 12 stages (stages I to XII) (Ahmed & de Rooij, 2009). These stages were thus used to evaluate spermatogenic cell distributions and arrangements in Fhad1−/− mice, revealing no abnormal changes in the seminiferous tubule architecture of these mice and no evidence of dysregulated spermatogenesis, with visible germ cells from spermatogonia to elongated spermatids. The phenotypes of these mice were comparable to those in the testes of Fhad1+/+ controls (Figs. 2D and 2E).

Given the observed expression of Fhad1 in spermatocytes, γH2AX staining of sections from Fhad1−/− mice was performed, revealing typical meiotic progression consistent with general meiosis remaining intact following the deletion of Fhad1 (Fig. 2F). Further, the ultrastructure of elongating spermatids was examined, with no difference observed between the groups (Fig. 2G). These data thus indicate that the absence of the testis-enriched Fhad1 gene did not adversely influence fertility or spermatogenesis in male mice.

Fhad1-/- mice exhibit reduced sperm motility

The impact of Fhad1 knockout on spermiogenesis was next assessed by comparing epididymal spermatozoa from male Fhad1+/+ and Fhad1-/- mice, with no differences observed when assessing H&E-stained epididymal sections from these two groups (Fig. 3A). H&E staining also revealed that the morphology of cauda epididymal spermatozoa was not markedly affected by the absence of FHAD1 (Figs. 3B and 3C). AC-TUBULIN signals showed no significant differences between Fhad1+/+ and Fhad1−/− spermatozoa, suggesting that the flagellar structure of Fhad1−/− spermatozoa was normal (Fig. 3D). Taken together, these results suggested that FHAD1 may be dispensable for the gross morphology of epididymal spermatozoa. TEM was next used to assess the ultrastructural characteristics of the flagella (Fig. 3E). Midpiece sections from both Fhad1+/+ and Fhad1−/− mice showed that the mitochondria formed an outer layer with a central axoneme surrounded by outer dense fibers (ODFs). Moreover, the axoneme, ODF, and fibrous sheath (FS) were clearly visible in samples from both Fhad1+/+ and Fhad1−/− mice, with wrapping of the membrane around the filaments in the principal piece. These intact structures suggested that Fhad1−/− sperm exhibit no abnormal morphology.

Figure 3 Fhad1−/−mice had reduced sperm motility.

(A) H&E-stained caput and cauda epididymis tissues. (B) Spermatozoa from Fhad1+/+ and Fhad1−/− mice were subjected to H&E staining. (C) Frequencies of sperm exhibiting normal morphological characteristics, n = 3. (D) AC-TUBULIN immunofluorescent detection in spermatozoa. (E) Electron microscopy images of Fhad1+/+ and Fhad1−/− spermatozoa. M, midpiece; P, principal piece; ODF, outer dense fibers; FS, fibrous sheath. (F) Cauda epididymal sperm contents from Fhad1+/+ and Fhad1−/− mice, n = 6, *P < 0.05. (G) Average motile sperm and (H) progressive sperm ratios of adult Fhad1+/+ and Fhad1−/− mice, n = 6, **P < 0.01.

To determine the sperm quality in Fhad1−/− males, we examined sperm function using a CASA analysis system (Castaneda et al., 2017). Statistically, there was no significant difference in sperm counts in the cauda epididymis between Fhad1+/+ and Fhad1−/− males (Fig. 3F). However, the total motility of Fhad1−/− sperm (69.4 ± 4.7), namely, the proportion of motile sperm to total sperm count, was significantly decreased compared to that in Fhad1+/+ mice (77.3 ± 5.6) (Fig. 3G). Similarly, the progressive motility of Fhad1−/− sperm (32.1 ± 4.8), namely, the proportion of sperm that move forward to the total sperm count, was also significantly lower than that of Fhad1+/+ sperm (40.48 ± 3.8) (Fig. 3H). This difference was slight and did not reduce the fertility of the mice under normal conditions. Nevertheless, it does suggest that FHAD1 may play a role in sperm motility.

Together, these data show that the loss of FHAD1 expression in mice may impair sperm motility, albeit not to a degree sufficient to have a detectable adverse impact on fertility.

FHAD1 deletion triggers the apoptosis of spermatocytes during the first wave of spermatogenesis

As spermatocytes express FHAD1, to further assess its role in these cells, spermatocytes were evaluated during the first wave of spermatogenesis. TUNEL staining of the testis tissues collected from Fhad1+/+ and Fhad1−/− mice at P16, when the first wave of spermatogenesis proceeds to the pachytene stage, was conducted. Relative to the testes of Fhad1+/+ mice, Fhad1−/− mice presented with a slight increase in apoptotic cell death (Figs. 4A–4C). These results suggested that FHAD1 plays a role in meiosis such that its deletion can induce apoptotic spermatocyte death during this process.

Figure 4 FHAD1 deficiency-induced germ cell apoptosis during the first spermatogenic wave.

(A) Paraffin-embedded testis sections from 16 Days Fhad1+/+ and Fhad1−/− mice stained with TUNEL. (B and C) Graphs show the number of TUNEL positive cells per tubule and the percentage of TUNEL positive tubules in the sections n = 8. (D) Immunofluorescence show the levels of γH2AX, a DNA damage marker. (E) γH2AX signals were counted in cells, ****P < 0.0001. (F) The percentages of γH2AX-positive cells per slide were shown, **P < 0.01. (G) Graphs showing the γH2AX signal intensity per cells, ***P < 0.001. Error bars, measn ± SEM, n > 50 cells for each group.

Lastly, FHAD1 was overexpressed in 293T cells which were then treated using cisplatin to promote DNA damage. Subsequent immunostaining for the DNA damage marker γH2AX revealed that overexpression of FHAD1 significantly inhibited DNA damage in these cells (Figs. 4D–4G), potentially supporting a role for FHAD1 as a modulator of DNA double-strand break (DSB) formation and repair during spermatocyte meiosis.

Discussion

In this study, the testis-enriched FHAD1 protein was found to play important functional roles as a regulator of meiotic cell division and sperm motility. Firstly, we identified FHAD1 as a highly expressed testicular protein, consistent with published studies, suggesting that FHAD1 played an important role in spermatogenesis and male fertility. While Fhad1−/− mice generated through a CRISPR/Cas9 approach exhibited normal fertility, a slight reduction in sperm motility was detected. Additional analyses of spermatogenesis in these Fhad1−/− mice revealed that FHAD1 deletion was associated with slightly higher spermatocyte apoptosis rates during the first wave of meiotic division. As such, these results provide the first evidence demonstrating that FHAD1 functions as a key protein involved in the regulation of spermatogenesis in vivo.

Previous studies in which testis-enriched genes were inactivated have successfully demonstrated the contributions of these genes to male infertility and the impairment of spermatogenesis in mice, underscoring their importance to reproductive outcomes. This approach has been used to evaluate many candidate testes-enriched genes to date, with many having been found to play detectable roles in the regulation of murine spermatogenesis (Castaneda et al., 2017; Gao et al., 2020; Hua et al., 2019; Shah et al., 2021; Shibuya et al., 2015; Yamase et al., 2019; Yang et al., 2021; Zhang et al., 2019), whereas others are dispensable in this context, including FANK1, Hspa1l, and Tex33 (Wang et al., 2020; Zhang et al., 2019; Zhu et al., 2020). It has been suggested that the proteins encoded by these genes may have redundant functions in spermatogenesis. However, other studies have revealed that while the loss of certain proteins does not result in infertility, they still serve as important regulators of spermatogenesis (Gadadhar et al., 2021). The advent of new gene knockout techniques has enabled detailed research efforts focused on identifying pathogenic genes associated with male infertility. These efforts have revealed that genes such as TEKT3 may be important causes of male infertility in the general population, even though their knockout in mice does not lead to infertility (Liu et al., 2023; Roy et al., 2009). These inconsistent phenotypes may be context- or species-specific. Overall, these past studies have key implications for efforts to identify the genes associated with the preservation or loss of male fertility.

Phosphorylation is the best-studied form of posttranslational modification in human spermatozoa (Porambo et al., 2012). Global phosphorylation profiling efforts in spermatozoa from a range of species support key roles for such phosphorylation in the regulation of key activities including spermatogenesis, maturation, motility, and capacitation (Serrano, Garcia-Marin & Bragado, 2022; Urizar-Arenaza et al., 2019). However, the precise molecular mechanisms that govern the motility of spermatozoa remain to be fully elucidated, as do the associated impacts on sperm quality. The study of these mechanisms will enable the more detailed interrogation of the biochemical factors that underlie poor semen quality and may reveal novel biomarkers associated with various forms of infertility-related conditions and pathologies. In the present study, the FHAD1 protein, which harbors an FHA domain that recognizes pSer/pThr residues, was selected as a focus for experimental analysis (Durocher et al., 2000). FHAD1 has been found to be highly enriched in the sperm flagellum, and has been presumed to play an important role in the spermiogenesis and motility (Pineau et al., 2019). The present data confirmed its expression, and the deletion-associated phenotypes suggested that it may serve as an important regulator of spermatogenesis. Future research efforts, however, will be critical to determine whether FHAD1 is a phosphorylation reader and what its substrate is, thereby providing a theoretical foundation for a comprehensive analysis of spermatogenesis-related phosphorylation activity.

The process by which DNA damage is recognized and then signaled to the DNA repair machineries can be considered as a bona fide signal transduction cascade. It is initiated by a signal (DNA damage) which is detected by a receptor or sensor. Then a proximal set of kinases would be activated, which then phosphorylate and regulate a number of protein effectors of the DNA damage responses (Zhou & Elledge, 2000). During this process, phosphorylation can serve as a biomarker of DNA damage. FHA-dependent peptide recognition plays a critical role in reading the phosphorylated sites and activating downstream DNA damage repair responses in this cascade (Durocher & Jackson, 2002).

In meiosis, after PRDM9-mediated changes to chromatin at recombination hotspots during the process of meiotic division, SPO11 catalyzes DSB formation, thereby activating a DNA damage response (Parvanov, Petkov & Paigen, 2010). This, in turn, triggers ATM kinase downstream H2AX phosphorylation, yielding an isoform referred to as γH2AX (Baudat et al., 2010). During the leptotene and zygotene stages, γH2AX is observed on autosomal chromatin and exhibits a widespread nuclear distribution before ultimately disappearing from autosomes after meiotic DSB repair such that it is only evident on the XY chromatin at the pachytene and diplotene stages, consistent with the meiotic inactivation of sex chromatin (Parvanov, Petkov & Paigen, 2010). During the initial wave of spermatogenesis, the absence of FHAD1 expression results in slightly elevated rates of apoptotic death consistent with it playing a functional role in the meiotic process. To test this possibility, cisplatin was used here to induce the formation of DSBs, revealing that FHAD1 overexpression was associated with a reduction in γH2AX levels. These findings support a model wherein FHAD1 can function as a phosphorylation site reader that can regulate the repair of DSBs. The data also suggest that FHAD1 may maintain fertility under extreme conditions, such as when people are treated with chemotherapy drugs. However, no highly specific FHAD1 antibody was available for the present study, and additional research will be necessary with a focus on immunofluorescent staining and the identification of proteins that interact with FHAD1 in order to better clarify its functional role.

In summary, while mice lacking FHAD1 expression were fertile, they did exhibit a reduction in sperm motility. The apoptotic death of spermatocytes lacking FHAD1 during the first phase of spermatogenesis suggests that this protein may regulate DSB formation and repair.

Supplemental Information

Supplemental Information 1 List of primer sequences.

Supplemental Information 2 Raw data for mRNA expression in tissues of Figure1A.

Supplemental Information 3 Raw data for mRNA expression in development stages of Figure1B.

Supplemental Information 4 Raw data for mRNA expression of Figure1C.

Supplemental Information 5 Raw data for fertility test of Figure 2A.

Supplemental Information 6 Raw data for testis-body weight of Figure2C.

Supplemental Information 7 Raw data for composition of different cell types Figure 2E.

Supplemental Information 8 Raw data for spermatozoa abnormality ratio of Figure 3C.

Supplemental Information 9 Raw data for spermatozoa concentration and motility of Figure 3F, 3G and 3H.

Supplemental Information 10 Raw data for TUNEL of Figure 4B and 4C.

Supplemental Information 11 Raw data for γH2AX of Figure 4E and 4F.

Supplemental Information 12 Author Checklist.

Supplemental Information 13 MIQE_checklist.

We would like to thank Prof. Rong Hua from Anhui Medical University for his discussion and support of this work.

Additional Information and Declarations

Competing Interests

Author Contributions

Animal Ethics

Data Availability

The authors declare that they have no competing interests.

Xi Zhang conceived and designed the experiments, performed the experiments, analyzed the data, authored or reviewed drafts of the article, and approved the final draft.

Jiangyang Xue performed the experiments, analyzed the data, authored or reviewed drafts of the article, and approved the final draft.

Shan Jiang performed the experiments, authored or reviewed drafts of the article, and approved the final draft.

Haoyu Zheng performed the experiments, analyzed the data, prepared figures and/or tables, and approved the final draft.

Chang Wang conceived and designed the experiments, performed the experiments, analyzed the data, prepared figures and/or tables, authored or reviewed drafts of the article, and approved the final draft.

The following information was supplied relating to ethical approvals (i.e., approving body and any reference numbers):

The Institutional Animal Care and Use Committee (IACUC) of Nanjing Medical University (Approval No. IACUC-1601117).

The following information was supplied regarding data availability:

The raw measurements are available in the Supplemental Files.

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
