# Peer review of "Forkhead-associated phosphopeptide binding domain 1 (FHAD1) deficiency impaired murine sperm motility"

_PeerJ, doi:10.7717/peerj.17142_

## Round 0.1 · original submission · Minor Revisions

Please address the critiques of both reviewers and amend the manuscript accordingly.

**Language Note:** The review process has identified that the English language must be improved. PeerJ can provide language editing services - please contact us at [email protected] for pricing (be sure to provide your manuscript number and title). Alternatively, you should make your own arrangements to improve the language quality and provide details in your response letter. – PeerJ Staff

Reviewer 1 ·

Basic reporting

• Authors used Clear, unambiguous, professional language throughout the manuscripts.
• Sufficient field background and context is missing. Therefore, this requires refinements prior to the acceptance.
For example,
The introduction section covers i) the general background of infertility, related genes and concluding with the knowledge gap; ii) CRISPR/Cas-mediated gene editing, background references that used this gene editing technologies to investigate fertility-related biological phenomena; iii) the importance of phosphorylation-mediated biological regulations; iv) importance of FHA domains in DNA repair-related processes and then they introduced the reason of choosing FHAD1 genes.
While the text presents a coherent progression of ideas, it lacks essential context and an elaborate explanation of how proteins containing FHA domains might participate in DNA repair mechanisms and, consequently, potentially influence spermatogenesis. Furthermore, it is unknown whether the presence of beta sheets in the forkhead-associated domain plays a role in the DNA repair process. Moreover, the nuclear localization signal of FHAD1 also corroborates the DNA-repair activities. Hence there is needed additional comments.
The authors rationalize the selection of the FHAD1 gene for this investigation, stating that its expression is notably elevated in the testis. Nevertheless, it exhibits noteworthy expressions in additional tissues, including lung tissue. Therefore, a cohesive elucidation of the protein's significance and potential association with spermatogenesis ought to be presented in an independent paragraph. Although it can be included in the introduction, it is more appropriate to place it in the initial paragraph of the results and discussion section.
The article structure, figures, and tables were professional and perfect. Raw data was shared, and it was effectively elucidated.
Although the manuscript contains enough relevant results to the hypotheses, more detailed experiments could be performed at the cellular level to elucidate the DNA double-stranded break (DSB). The authors employed FHAD1 protein overexpression in HEK293T cells to assess its role in DNA damage repair. They utilized fluorescence microscope to assess their findings. Alternatively, in the same study, an additional positive control (knockdown of FHAD1 in HEK293T cells) can be employed. While fluorescence imaging provides the desired outcome, flow cytometry can be used to get quantitative results.
Please find the additional comments below.

Experimental design

• Original primary research within the Aims and Scope of the journal.
• Research question well defined, relevant & meaningful. The findings of the study indicate that FHAD1-/- mice exhibited no alterations in fertility or testicular morphology. The absence of FHAD1, on the other hand, results in a notable decrease in sperm motility and apoptosis during the initial phase of spermatogenesis. FHAD1, like different other proteins implicated in the process of spermatogenesis, exhibits a substantial coiled-coil motif in addition to its forkhead-associated domains. Thus, it is predicted to have a function related to DNA transcription. However, due to the lack of genetic and chemical tools, the exact role of this protein in spermatogenesis remained unexplored. By investigating the possible role of the FHAD1 protein in mouse spermatogenesis, this study attempts to fill this knowledge gap using CRISPR/Cas knock-out technology. Thus, the manuscript illuminated the novel function of FHAD1 in mouse spermatogenesis.
• Rigorous investigation performed to a high technical & ethical standard.
• Methods not described with sufficient details. For example,
1. Line 233: “Further ultrastructure of elongated spermatids was detected…. (Fig. 2G).” and Line 241: “TEM was next used to assess… flagella (Figure 3E).” There is no mention of TEM sample preparation in the methods section. Please include one short para for TEM sample preparation, which will be helpful for readers to follow up.
2. The results illustrated in Figure 3F-3H serve as the foundation for the main conclusion that can be deduced from this manuscript. It merits a comprehensive elucidation within the results and discussion section. The readers will be interested to learn how CAS-based analysis is implemented and how the results of this experiment are evaluated to arrive at a conclusion.
While the authors noted and presented these points in Line 237: "Fhad1-/- mice exhibit reduced sperm motility" section, they could be elaborated upon in a distinct section that comprises the following: the experimental design followed by appropriate citations; the obtained results; and an explanation of Fig. 3F-H. The method description, analysis and result interpretation for CAS-based sperm motility analysis requires attention.

Validity of the findings

• Impact and novelty have been assessed but need more detailed correlation with currently existing literature.

Additional comments

The current manuscript by Wang and coworkers reports the function of Forkhead-associated phosphopeptide binding domain 1 (FHAD1) in the regulation of motility of murine spermatozoa. According to their result, the FHAD1 gene also appears to be involved in a DNA repair-related function during spermatogenesis.
The authors employed a CRISPR/Cas9-mediated gene-knockout strategy to generate a mouse model deficient in FHAD1 (referred to as FHAD1-/-), which was subsequently used to investigate spermatogenesis and fertility. The success of the knockout experiments was first verified through qPCR analysis. Thereafter, the authors used immunofluorescence, histological staining, and transmission electron microscopy (TEM) to visualize any change in testis phenotypes for FHAD1-/- mice. Then, computer-assisted methodologies were employed to analyze sperm concentration and motility. Lastly, they used different chemical assays to validate the possible role of FHAD1 in protecting spermatocytes from apoptotic cell death at the first wave of spermatogenesis.
The findings of the study indicate that FHAD1-/- mice exhibited no alterations in fertility or testicular morphology. The absence of FHAD1, on the other hand, results in a notable decrease in sperm motility and apoptosis during the initial phase of spermatogenesis. FHAD1, like different other proteins implicated in the process of spermatogenesis, exhibits a substantial coiled-coil motif in addition to its forkhead-associated domains. Thus, it is predicted to have a function related to DNA transcription. However, due to the lack of genetic and chemical tools, the exact role of this protein in spermatogenesis remained unexplored. By investigating the possible role of the FHAD1 protein in mouse spermatogenesis, this study attempts to fill this knowledge gap using CRISPR/Cas knock-out technology. Thus, the manuscript illuminated the novel function of FHAD1 in mouse spermatogenesis.
While the authors demonstrate thorough experimental planning and data analysis, the following points should be addressed prior to acceptance.
1. Line 195: “To begin exploring the possible functional role …… whereas it was largely undetectable in other tissue types”. Please include a reference for this statement. While in the author’s result (Figure 1A), Te (Testis) shows the highest expression level of Fhad1, Lu (lung) also shows a little expression of Fhad1, which is also consistent with previous reports by NCBI’s EST profile. Hence, it corroborates the previous knowledge. Thus, it deserves proper citation and detailed explanation.
2. The previous report shows that FHAD1 also exhibits a distinct flagellar signal in late spermatids (DOI: 10.1021/acs.jproteome.9b00351). Furthermore, their flagella-restricted localization in spermatids implies that they may be involved in the structure and functions of the flagellum. Nonetheless, this report merely hypothesized FHAD1's role during spermatogenesis. This manuscript highlighted a significant role that FHAD1 plays in sperm motility. Hence, it is imperative to cite this and related references properly, and establishing a suitable correlation between previous and current works pertaining to the function of FHAD1 could contribute to the progression of the ideas.
3. Line 143: “A modified version of previously published approach was used…. cells”; please include the reference of this previously published research.
4. Line 202: “When fluorescence-activated cell sorting was used to purify germ cells…both haploid and tetraploid cells, consistent with the presence of spermatids and spermatocytes”;
a) Spermatids (formed by the meiotic division of secondary spermatocyte, which differentiates into sperm cells
are normally haploid cells, while spermatocytes are normally considered diploid cells. However, authors represented spermatocytes as tetraploid cells.
b) In the process of cell sorting, fluorescence-based methods were used to isolate different ploidy cells, and those were subjected to qPCR analysis for Fhad1 proteins. But according to the sentence, it is not clear. A more refined sentence could be easy to follow up.
5. Line 233: “Further ultrastructure of elongated spermatids was detected…. (Fig. 2G).” and Line 241: “TEM was next used to assess… flagella (Figure 3E).” There is no mention of TEM sample preparation in the methods section. Please include one short para for TEM sample preparation, which will be helpful for readers to follow up.
6. Line 238: “The impact of Fhad1 knockout on spermatogenesis was next assessed by comparing the morphology of cauda epididymal spermatozoa… between these groups (Fig. 3A-D)”; one or two sentences which include more explanation on these results could be helpful to follow up. It represents a beautiful result, which shows that the structure and morphology of cauda epididymal spermatozoa have not been affected by the knockout of the Fhad1 gene. Subsequent investigations in the field will find this result helpful. Hence, as the authors did for Figure 3E, a convincing explanation is required.
7. The results illustrated in Figure 3F-3H serve as the foundation for the main conclusion that can be deduced from this manuscript. It merits a comprehensive elucidation within the results and discussion section. The readers will be interested to learn how CAS-based analysis is implemented and how the results of this experiment are evaluated to arrive at a conclusion.
While the authors noted and presented these points in Line 237: "Fhad1-/- mice exhibit reduced sperm motility" section, they could be elaborated upon in a distinct section that comprises the following: the experimental design followed by appropriate citations; the obtained results; and an explanation of Fig. 3F-H.
8. Line 279: “ used to evaluate mant candidate testes-enriched…….found to play”, it should be many candidate testes.
9. Figure 2F, it is ʏ/H2AXHoechst in the manuscript, which should be ʏH2AX/Hoechst.
10. Figure 4A, it is TUNLE/Hoechst in the manuscript, which should be TUNEL/Hoechst. These are just two examples. Authors should check each figure for such mistakes.
11. Figure 1B and 1C, Relative mRNA expression, please mention that mRNA expression for Fhad1.

Reviewer 2 ·

Basic reporting

1) The manuscript requires extensive re-writing and grammar check.
2) The material and methods section is developed poorly and lacks many details such as information on sgRNA used for CRISPR.
3) Reference needed in lines 55, 83
4). Define SPF conditions.
5) Line 95- no experimental interventions were carried out. This line is very vague and redundant.

Experimental design

1) Was cervical dislocation done without prior anesthesia?
2) It would be beneficial to label or add an arrow for axoneme, ODF, and FS.
3) Explain the difference between total vs progressive motility and how they were calculated.
4) FHA domains are extensively involved in transcription but no experiment was conducted to show that overall transcription or translation is not effected.

Validity of the findings

1) Although the authors suggest that sperm count was similar, in Figure 3 F it looks substantially decreased.
2) What is the biological relevance of increased apoptosis in spermatocytes when the overall sperm count is same, although looks less in Figure 3?
3) Authors failed to link their findings. When overall fertility is not impacted, then how does FHAD1 play a significant role in human biology?

---

## Round 0.2 · accepted · Accept

All remaining concerns of the reviewers were adequately addressed, and the revised manuscript is acceptable now.

Reviewer 1 ·

Basic reporting

No comment

Experimental design

No comment

Validity of the findings

No comment

Reviewer 2 ·

Basic reporting

The authors addressed my concerns.

Experimental design

The authors addressed my concerns.

Validity of the findings

The authors addressed my concerns.

Additional comments

The authors addressed my concerns.